# Magnetic Resonance Imaging Liver Segmentation Protocol Enables More Consistent and Robust Annotations, Paving the Way for Advanced Computer-Assisted Analysis

**DOI:** 10.3390/diagnostics14242785

**Published:** 2024-12-11

**Authors:** Patrick Jeltsch, Killian Monnin, Mario Jreige, Lucia Fernandes-Mendes, Raphaël Girardet, Clarisse Dromain, Jonas Richiardi, Naik Vietti-Violi

**Affiliations:** 1Department of Radiology and Interventional Radiology, Lausanne University Hospital, Lausanne University, 1015 Lausanne, Switzerland; patrick.jeltsch@chuv.ch (P.J.); killian.monnin@chuv.ch (K.M.); mario.jreige@chuv.ch (M.J.); lucia.fernandes-mendes@chuv.ch (L.F.-M.); clarisse.dromain@chuv.ch (C.D.); jonas.richiardi@chuv.ch (J.R.); 2Department of Radiology, South Metropolitan Health Service, Murdoch, WA 6150, Australia; raphael.girardet@health.wa.gov.au

**Keywords:** segmentation, liver, MRI, computer-assisted analysis, radiomics, deep learning

## Abstract

Background/Objectives: Recent advancements in artificial intelligence (AI) have spurred interest in developing computer-assisted analysis for imaging examinations. However, the lack of high-quality datasets remains a significant bottleneck. Labeling instructions are critical for improving dataset quality but are often lacking. This study aimed to establish a liver MRI segmentation protocol and assess its impact on annotation quality and inter-reader agreement. Methods: This retrospective study included 20 patients with chronic liver disease. Manual liver segmentations were performed by a radiologist in training and a radiology technician on T2-weighted imaging (wi) and T1wi at the portal venous phase. Based on the inter-reader discrepancies identified after the first segmentation round, a segmentation protocol was established, guiding the second round of segmentation, resulting in a total of 160 segmentations. The Dice Similarity Coefficient (DSC) assessed inter-reader agreement pre- and post-protocol, with a Wilcoxon signed-rank test for per-volume analysis and an Aligned-Rank Transform (ART) for repeated measures analyses of variance (ANOVA) for per-slice analysis. Slice selection at extreme cranial or caudal liver positions was evaluated using the McNemar test. Results: The per-volume DSC significantly increased after protocol implementation for both T2wi (*p* < 0.001) and T1wi (*p* = 0.03). Per-slice DSC also improved significantly for both T2wi and T1wi (*p* < 0.001). The protocol reduced the number of liver segmentations with a non-annotated slice on T1wi (*p* = 0.04), but the change was not significant on T2wi (*p* = 0.16). Conclusions: Establishing a liver MRI segmentation protocol improves annotation robustness and reproducibility, paving the way for advanced computer-assisted analysis. Moreover, segmentation protocols could be extended to other organs and lesions and incorporated into guidelines, thereby expanding the potential applications of AI in daily clinical practice.

## 1. Introduction

Cirrhosis is increasing worldwide and stands as a significant contributor to mortality, primarily due to the increased risk of hepatocellular carcinoma (HCC) in affected patients [1]. Indeed, HCC is the second leading cause of cancer-related death worldwide and the fifth in Europe [2,3]. MRI is considered the gold standard for HCC diagnosis [4]. The Liver Imaging Reporting & Data System (LI-RADS) algorithm permits definitive diagnosis of HCC without pathologic confirmation when applied in patients with cirrhosis [5]; however, it remains subject to interpretation bias. This has sparked interest in developing AI-based computer-assisted analysis for imaging examinations [6]. Two prominent approaches, radiomics and deep learning (DL), are currently the most actively investigated techniques [7,8,9]. These two techniques have demonstrated their utility in applications such as characterizing focal hepatic lesions, staging liver fibrosis, and identifying portal hypertension [10]. They both have an exceptional potential in assisting radiologists by enabling earlier disease detection, more precise disease classification, and providing additional prognostic information to guide clinical management [11].

Robust and reliable application of these techniques relies on automated image processing pipelines, where one of the early steps is to locate the liver and isolate it from surrounding organs, a process known as segmentation. More broadly, the scarcity of high-quality annotated datasets represents a critical bottleneck for radiomics features extraction and DL algorithms performance [12,13]. Annotation-related problems may be particularly relevant in the field of biomedical image analysis, where data are typically sparse, inter-reader variability is naturally high, labeling ambiguities occur, and medical experts have their individual styles of annotations [14]. Segmentation of the regions of interest can be performed using three different approaches: manual, semi-automatic, and automatic. Manual segmentation is the most reliable and has the advantage of higher accuracy, although it is time-consuming and subject to inter-reader variability [15]. Automatic segmentation aims to automatically identify the regions of interest using computer algorithms, and semi-automatic segmentation combines automated processes with manual corrections as needed [16]. These last two methods offer the benefit of faster processing times. However, most proposed automated liver segmentation techniques were developed using small datasets, limiting their ability to deliver consistent and generalizable performance on external datasets [17,18].

While numerous automated liver segmentation algorithms have been developed on CT [19,20,21], fewer studies have focused on MRI liver segmentation. Several automatic MRI liver segmentation methods based on convolutional neural networks have been proposed [22,23], but manual tracing is still regarded as the gold standard and remains essential for the advancement of automatic MRI liver segmentation algorithms [24,25]. Liver MRI segmentation presents a significant challenge due to variations in size, respiratory motion, relative organ position, and phase acquisition at different points in the respiratory cycle, which complicate the segmentation process [26]. Additionally, fibrosis processes in liver cirrhosis can significantly alter the organ’s morphology [27]. Currently, there is no consensus on the method of liver segmentation for computer-assisted analysis, especially with MRI. However, labeling instructions and exemplary images have proved to be crucial for improving the quality of annotated datasets [14]. Despite their apparent importance, earlier research revealed that labeling instructions are typically not provided [28].

The purpose of this study is to establish an illustrative liver MRI segmentation protocol and evaluate its potential to enhance the quality of annotated datasets through assessment by two professional annotators.

## 2. Materials and Methods

### 2.1. Patient Selection

This retrospective single-center study included 20 patients (M/F: 15/5, mean age: 54 y.o.), selected from a pre-existing database of liver MRIs collected in a cohort of patients with chronic liver disease, who underwent extracellular gadolinium-based contrast media liver MRI, as previously published [29]. The patient selection criteria were as follows: random selection of 10 patients with cirrhosis and 10 patients without cirrhosis, all of whom were part of the previously mentioned publication. In cases where liver MRI scans were affected by significant motion artifacts, another patient was randomly selected. The study was approved by the local Institutional Review Board (ID CER-VD 2020-00680), which waived the need for signed informed consent.

### 2.2. Liver Segmentations and Protocol

All liver MRIs were conducted using a 3-T MRI system (Magnetom Skyra, Siemens Healthineers, Erlangen, Germany) with a standard liver-dedicated protocol. Segmentations were performed on Fat-Saturated (FS) T2wi, and T1wi in the portal venous phase. Liver segmentations were performed manually using the medical software Mint Lesion™ (version 3.9.1, Heidelberg, Germany) by two annotators: a radiologist in training (P.J.) and a radiology technician (L.F.M.). Two rounds of segmentation were conducted. On the first round, minimal instructions were provided to the annotators. All segmentations were reviewed and corrected in consensus among two expert radiologists (C.D. and N.V.V., with 25 and 8 years of expertise in abdominal imaging, respectively) and a radiologist specializing in radiomics data extraction (M.J., with 8 years of expertise in this domain) in order to develop a segmentation protocol. To reduce discrepancies after the first round, specific regions of the liver where the differences between the two annotators were most significant were analyzed using the corrected segmentations by the experts as the reference standard. Key areas of discrepancy included the hepatic hilum, vascular structures, and ligamentous regions. Additionally, clear instructions were provided to the annotators for specific cases, such as those involving multi-part liver parenchyma, respiratory motion artifacts, and ascites. The segmentation protocol was developed to standardize the process and enhance consistency between annotators. The second round was performed on the same liver MRI dataset, adhering to the protocol. In order to avoid recall bias, the second round was performed at least six weeks after the first round. The segmentation process is illustrated in Figure 1.

The segmentation protocol provides clear guidance for accurately segmenting the liver parenchyma, including the importance of appropriate windowing and magnification. It also details how to exclude extra-hepatic structures, such as the hepatic hilum, vessels, and ligaments. Finally, it provides recommendations in cases of multi-part liver, respiratory motion artifacts, and ascites. A detailed description of the liver segmentation protocol is provided in Appendix B.

### 2.3. Statistical Analysis

The Dice Similarity Coefficient (DSC) and the Hausdorff Distance (HD) were used to evaluate the inter-reader agreement pre- and post-protocol implementation on a per-volume (whole liver) basis. Statistical significance was assessed using the Wilcoxon signed-rank test to determine if metrics were significantly improved with the protocol. A subgroup analysis was conducted to assess liver segmentation correlation for patients with and without cirrhosis on a per-slice basis, using the Aligned-Rank Transform (ART) for repeated measures analyses of variance (ANOVA). Additionally, inter-reader agreement in selecting slices at the extreme cranial or caudal positions of the liver was evaluated pre- and post-protocol and assessed with the McNemar test. The level of significance for these statistical tests was set to 0.05. The metrics were calculated using the Python package seg-metrics v1.1.3 [30]. The Wilcoxon signed-rank test was performed with SciPy v1.10.1 [31], the ART for repeated measures ANOVA was applied using the ARTool package in the R software v4.2.2 [32], and the McNemar test utilized the library Statsmodels v0.13.5 [33].

## 3. Results

### 3.1. Patient Population

The majority of patients were Caucasian. The most frequent causes of liver disease were alcohol for patients with cirrhosis and hepatitis B virus for patients without cirrhosis. Patient demographics and clinical characteristics are summarized in Table 1.

### 3.2. Liver Segmentation Correlation

The per-volume inter-reader agreement evaluation (Table 2) exhibited a DSC improvement after protocol implementation from 0.944 ± 0.013 to 0.957 ± 0.008 on T2wi and from 0.953 ± 0.011 to 0.957 ± 0.009 on T1wi, both with a statistically significant difference (*p* < 0.001 and *p* = 0.03 on T2wi and T1wi, respectively). The HD reduced after protocol implementation from 24.47 ± 13.01 to 19.94 ± 5.38 on T2wi and from 21.85 ± 11.15 to 16.40 ± 5.68 on T1wi, with a statistically significant difference on T1wi (*p* = 0.048) but not on T2wi (*p* = 0.216).

Additionally, the protocol implementation resulted in a reduction in the number of liver segmentations with a non-annotated slice by at least one annotator at the extreme cranial or caudal slice of the liver, reducing from 17/20 to 11/20 segmentations on T2wi and from 20/20 to 12/20 on T1wi, with a statistically significant difference on T1wi (*p* = 0.036) but not on T2wi (*p* = 0.168).

In the subgroup analysis (Table 3, Appendix A and Appendix A) conducted to assess liver segmentation correlation for patients with and without cirrhosis on a per-slice basis, there was an average of 38.9 ± 2.8 slices per patient on T2wi and 78.8 ± 8.5 slices on T1wi. The protocol implementation showed an improvement in the median DSC per slice of patients with cirrhosis from 0.935 to 0.950 on T2wi and from 0.944 to 0.952 on T1wi. This improvement was more pronounced at the 10th (0.744 to 0.867 on T2wi and 0.840 to 0.862 on T1wi), 15th (0.825 to 0.898 on T2wi and 0.896 to 0.912 on T1wi), and 20th (0.864 to 0.913 on T2wi and 0.902 to 0.919 on T1wi) percentiles, representing the DSC on slices with most discrepancies.

The ART for repeated measures ANOVA (Table 4) highlights that the DSC is significantly influenced slice-wise by the protocol (*p* < 0.001 on T2wi and T1wi) and patient-wise by the presence of cirrhosis (*p* = 0.003 on T2wi and *p* < 0.001 on T1wi). The significant interaction of these two effects on both modalities (*p* < 0.001 on T2wi and T1wi) suggests that the combined effects of applying a protocol for annotating patient slices with cirrhosis significantly impact the inter-agreement of annotations.

### 3.3. Comparison of Annotations

This section illustrates the segmentation process, presenting MRI liver segmentation slices created by the annotators. Examples are provided in Figure 2, Figure 3, Figure 4 and Figure 5.

## 4. Discussion

In the present study, we established an illustrative liver MRI segmentation protocol that reduces inter-annotator variability and enhances the quality of annotated datasets. Annotation-related problems are particularly relevant in the field of biomedical image analysis, where labeling ambiguities occur, and medical experts have their individual styles of annotations [14]. During the creation of this segmentation protocol, we identified labeling ambiguities and pinpointed liver regions that are particularly challenging to label. Consequently, clear labeling instructions were provided to annotators, specifying actions such as excluding the hepatic hilum and identifying the vessels or ligaments to be segmented. Specifically regarding MRI, factors such as respiratory motion and phase acquisition at different points in the respiratory cycle significantly complicate the segmentation process. As detailed in the protocol, we elected to leave a larger margin at the edge of the hepatic contour to minimize the impact of these artifacts. Furthermore, chronic liver disease can markedly alter liver morphology and, in some cases, lead to the development of ascites. The protocol emphasizes the importance of windowing to delineate the hepatic contour and provides guidance for cases with ascites.

In the context of liver segmentation for pre-operative volumetry assessment, extreme precision is not warranted. Several semi-automatic and automatic liver segmentation models have been shown to yield acceptable measurements when compared to manual segmentation, as the total liver volume presents limited variation according to the technique used [34,35,36]. Conversely, in the domain of radiomics and DL, high-quality segmentation is critical, and manual tracing is still regarded as the gold standard [24]. Indeed, as evidenced by Poirot et al., in brain imaging, subtle variations in volume segmentation have a high impact on radiomics model robustness [37]. Consequently, segmentation is one of the early steps that require improvement at the beginning of the radiomics and deep learning algorithm pipelines. When considering whole-liver volume segmentation, precise segmentation of the liver parenchyma is essential to avoid incorporating adjacent tissue that might distort the extracted features. While the variation in radiomics features for HCC segmentation has been studied [38,39], to our knowledge, the impact of whole-liver MRI segmentations on computer-assisted analysis has not yet been investigated. However, high-quality segmentation is a prior step that could reduce variation in advanced computer analysis, making this protocol a potential first solution.

The implementation of the protocol significantly enhanced the metrics measured on a per-volume basis, leading to an increase in the DSC on both T2wi and T1wi phases. This improvement indicates more consistent and robust annotations, reducing inter-reader variability. Similarly, the significant decrease in the HD suggests tighter delineations of the liver. The more pronounced reduction in the HD on T1wi compared to T2wi may be attributed to the T2wi slice thickness (6 mm), which was double the T1wi slice thickness (3 mm), thereby reducing discrepancies. The evaluation of these metrics on a per-slice basis revealed a significant improvement in DSC, confirming the enhanced consistency of the annotators in contouring the liver. The use of a protocol also significantly improved the consistency of the annotators in selecting the extreme cranial and caudal slices of the liver in T1wi. These metrics improvements are particularly relevant for advanced computer-assisted analysis. Finally, in the subgroup analysis, the implementation of the protocol showed higher correlation improvement in patients with cirrhosis compared with patients without cirrhosis. These findings highlight the importance of using a segmentation protocol in case of cirrhosis, where anatomical alterations due to fibrosis increase annotation variations.

In comparison to another study evaluating the inter-annotator agreement in liver segmentation on T1wi, the DSC we obtained before protocol implementation (0.953 ± 0.011) was close to the reported DSC (0.95 ± 0.01) [40]. This study compared two board-certified abdominal radiologists while we compared a radiologist in training and a radiology technician. The initial inter-reader agreement in our study was further improved by the use of a segmentation protocol. This study focuses on delineating the contours of a shape (liver) rather than segmenting liver lesions, which requires the expertise of a specialized radiologist to detect and characterize lesions. We believe that this relatively straightforward task can be performed by radiologists in training and radiology technicians who have a certain level of anatomical knowledge after having received a well-defined and illustrative segmentation protocol. Such an approach could facilitate the segmentation process and reduce tasks that require the expertise of a specialized radiologist. The same was evidenced by Suman et al., who evaluated the performance of trained radiology technicians in volumetric pancreas segmentation on CT using a standard operating procedure that included video and images materials, and demonstrated that trained radiology technicians could achieve reasonable accuracy in the pancreas [41]. To our knowledge, the use of a liver segmentation protocol to train radiologists and radiology technicians in producing high-quality annotated datasets has not yet been explored. This approach could potentially accelerate the generation of datasets available.

As emerging noninvasive tools, radiomics and DL have shown promising performance in assisting radiologists [42]. Despite the rapid growth of AI, its impact on daily clinical practice remains limited due to a lack of reproducibility across studies [43]. To overcome this challenge, guidelines, standardization, and open-access datasets are essential for improving robustness in radiomics features extraction and addressing the reproducibility issues in DL. Stanzione et al. have proposed a CheckList for EvaluAtion of Radiomics research (CLEAR) as a standardization tool to facilitate the repeatability and reproducibility of radiomics studies [44]. To our knowledge, no study has proposed a standardized protocol for accurate manual segmentation. Although Macdonald et al. mentioned the use of a standard written protocol for their publicly available dataset of liver MRI segmentation, this protocol is only briefly mentioned and not provided in detail [45].

The present study has limitations. The main limitation is the small size of the patient cohort, although 160 segmentations were performed. However, the cohort was intentionally designed to consider different liver morphotypes by including patients with and without cirrhosis. Despite the limited number of patients, we observed a positive impact attributed to the segmentation protocol. Moreover, the present study was designed by following a previous study comparing two board-certified radiologists [40], which is why we chose to include only two annotators. However, beyond the analysis of inter-annotator variability itself, the primary finding demonstrated in our study is that implementing a segmentation protocol significantly improves annotation quality, which is essential for advanced computer-assisted analysis. Additionally, not all MRI sequences were segmented, but only the most commonly used sequences, for radiomics and DL analysis, including T2wi and T1wi at the portal venous phase. Finally, as the study was conducted in a single center on a European population, the findings may not generalize to other populations or to those imaged using different devices. Further studies are needed to evaluate the effect of the protocol on a larger-scale cohort and to assess its impact on advanced computer-assisted analysis, including radiomics feature extraction.

## 5. Conclusions

Our study demonstrates the added value of implementing a standardized and illustrative protocol to enhance the robustness and reproducibility of liver segmentation. Moreover, this protocol could reduce the need for specialized radiologists in the annotation process, potentially accelerating the creation of annotated databases. Moving forward, segmentation protocols should be extended to other organs and lesions and incorporated into guidelines for radiomics and DL pipelines, thereby expanding the potential applications of AI in daily clinical practice.

## Figures and Tables

**Figure 1 diagnostics-14-02785-f001:**
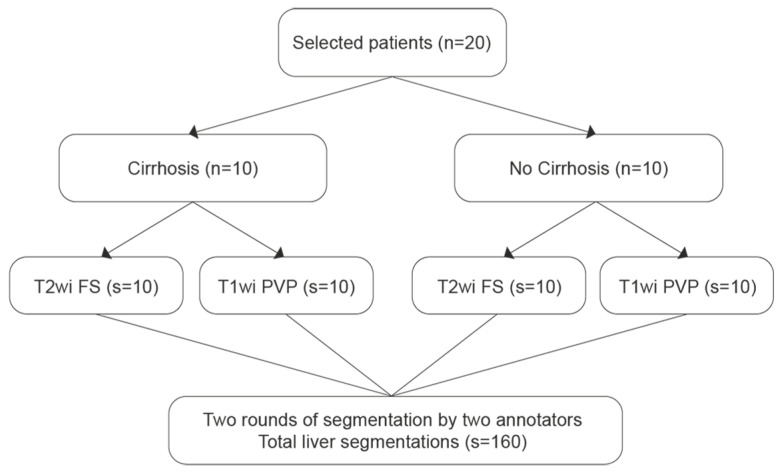
Liver segmentations process. Abbreviations: *n*—number of patients; wi—weighted imaging; FS—Fat Saturated; s—number of segmentations; PVP—portal venous phase.

**Figure 2 diagnostics-14-02785-f002:**
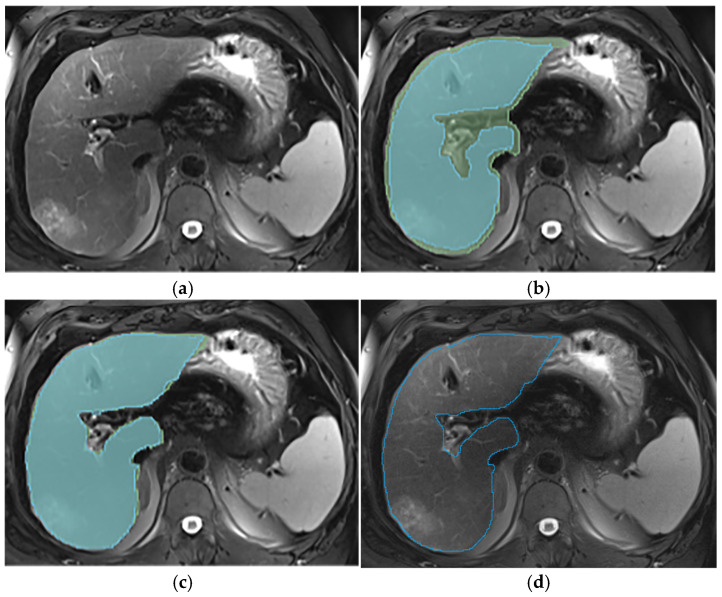
Examples of liver segmentations on T2wi (**a**–**d**) illustrated in blue (first annotator) and green (second annotator) before (**b**) and after (**c**) protocol implementation; (**d**) illustrates the expert segmentation, which was considered as the reference. This figure illustrates the importance of excluding the hepatic hilum and the variation in margin taken by the annotators at the border of the liver. Before protocol implementation (**b**), each annotator segmented the liver with their own style; one included the hilum (green segmentation), while the other did not (blue segmentation). Additionally, one annotator (blue segmentation) took a larger margin than the other at the border of the liver. After protocol implementation (**c**), these differences were lessened, improving the precision of the liver delineation and the inter-reader agreement.

**Figure 3 diagnostics-14-02785-f003:**
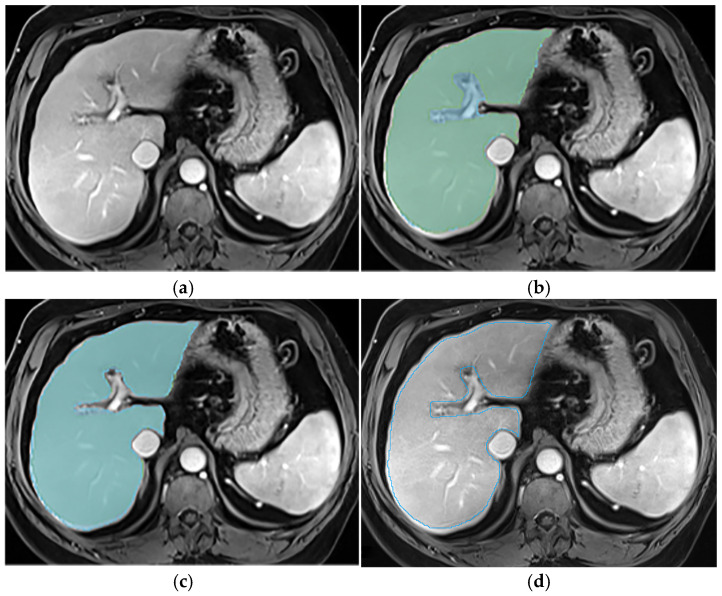
Examples of liver segmentations on T1wi (**a**–**d**) before (**b**) and after (**c**) protocol implementation; (**d**) illustrates the expert segmentation, which was considered as the reference. The portal vein was not always excluded from the liver contour before the use of the protocol (**b**). One annotator included it (blue segmentation), leading to potential overestimation of the liver volume, while the other did not (green segmentation). The implementation of the protocol ensured its exclusion in the annotations (**c**).

**Figure 4 diagnostics-14-02785-f004:**
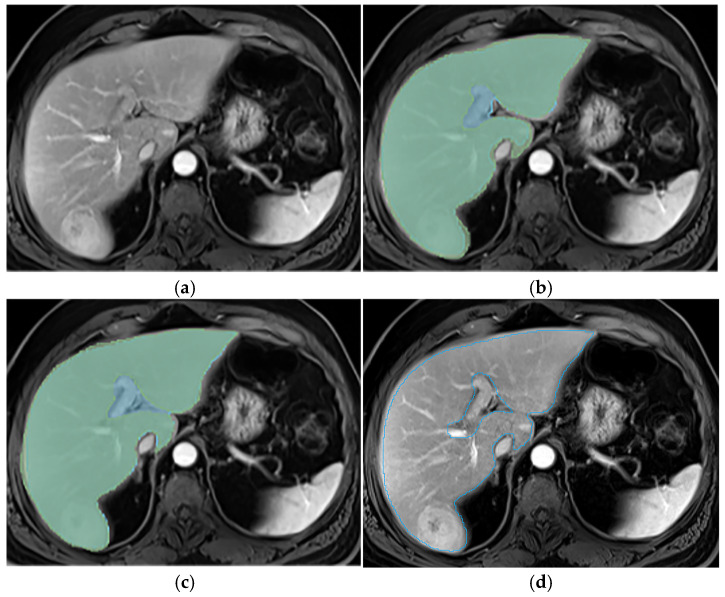
Examples of liver segmentations on T1wi (**a**–**d**) before (**b**) and after (**c**) protocol implementation; (**d**) illustrates the expert segmentation, which was considered as the reference. Sometimes, the use of the protocol did not improve the inter-reader agreement, since one annotator (blue segmentation) still included the portal vein while the other did not.

**Figure 5 diagnostics-14-02785-f005:**
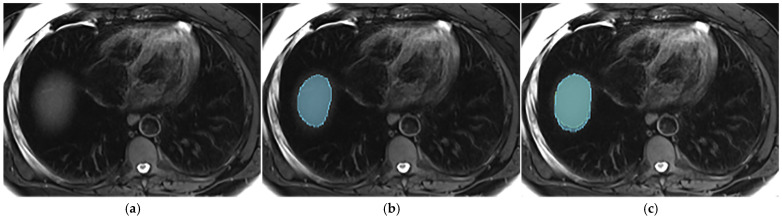
Examples of annotations on T2wi (**a**) with a missing extreme cranial slice from one annotator (green segmentation not visible) before the use of the protocol (**b**), while it was segmented by both annotators (blue and green segmentations) after protocol implementation (**c**).

**Table 1 diagnostics-14-02785-t001:** Baseline patient characteristics.

Characteristic	Study Cohort (*n* = 20)
Sex (M/F)	15/5
Age (Median, Range)	54, 29–79
Ethnicity	
Caucasian	13 (65%)
Asian	4 (20%)
African	3 (15%)
Liver Disease Etiology	
HBV	7 (35%)
Alcohol Consumption	6 (30%)
HCV	4 (20%)
NASH	3 (15%)
Cirrhosis	
Yes	10 (50%)
No	10 (50%)
Child-Pugh Class (If Cirrhosis)	
A	9 (90%)
B	1 (10%)

Abbreviations: *n*—number of patients; M—male; F—female; HBV—hepatitis B virus; HCV—hepatitis C virus; NASH—nonalcoholic steatohepatitis.

**Table 2 diagnostics-14-02785-t002:** Per-volume comparisons of metrics before and after protocol.

	Before Protocol (±SD)	After Protocol (±SD)	Wilcoxon Signed-Rank Test *p*-Value
T2wi	DSC = 0.944 ± 0.013	DSC = 0.957 ± 0.008	<0.001
	HD = 24.47 ± 13.01	HD = 19.94 ± 5.38	0.216
T1wi	DSC = 0.953 ± 0.011	DSC = 0.957 ± 0.009	0.03
	HD = 21.85 ± 11.15	HD = 16.40 ± 5.68	0.048

Abbreviations: SD—standard deviation; wi—weighted imaging; DSC—Dice Similarity Coefficient; HD—Hausdorff Distance.

**Table 3 diagnostics-14-02785-t003:** DSC per-slice comparisons of metrics in the subgroup analysis for patients with and without cirrhosis.

	Cirrhosis Status	Protocol	Mean	Median	q10	q15	q20
T2wi	Cirrhosis	Before	0.838	0.935	0.744	0.825	0.864
	Cirrhosis	After	0.905	0.950	0.867	0.898	0.913
	Without cirrhosis	Before	0.909	0.952	0.873	0.897	0.916
	Without cirrhosis	After	0.928	0.956	0.879	0.909	0.929
T1wi	Cirrhosis	Before	0.881	0.944	0.840	0.882	0.902
	Cirrhosis	After	0.913	0.952	0.862	0.896	0.919
	Without cirrhosis	Before	0.918	0.960	0.879	0.912	0.927
	Without cirrhosis	After	0.929	0.960	0.885	0.910	0.924

Abbreviations: wi—weighted imaging; q10—10th percentile; q15—15th percentile; q20—20th percentile.

**Table 4 diagnostics-14-02785-t004:** Aligned-Rank Transform (ART) for repeated measures analyses of variance (ANOVA).

	Factor	Error	F-Value	*p*-Value
T2wi	Protocol	Patient	1.664	0.215
	Protocol	Slice	38.233	<0.001
	Cirrhosis	Patient	11.797	0.003
	Protocol and Cirrhosis	Patient	0.767	0.394
	Protocol and Cirrhosis	Slice	14.262	<0.001
T1wi	Protocol	Patient	15.983	0.001
	Protocol	Slice	18.008	<0.001
	Cirrhosis	Patient	19.563	<0.001
	Protocol and Cirrhosis	Patient	0.093	0.764
	Protocol and Cirrhosis	Slice	14.602	<0.001

Abbreviations: wi—weighted imaging.

## Data Availability

Data are contained within the article and Appendix A.

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
