# Peer review of "Magnetic Resonance Imaging Liver Segmentation Protocol Enables More Consistent and Robust Annotations, Paving the Way for Advanced Computer-Assisted Analysis"

_diagnostics, 2024, doi:10.3390/diagnostics14242785_

Round 1
Reviewer 1 Report
Comments and Suggestions for Authors
I have the following comments:
1) While multiple slices were segmented (making up a total of 160 segmentations), 20 patients (of whom 10 with cirrhosis and 10 without cirrhosis) is a very limited sample size, posing a risk of the study being underpowered - this could especially affect statistically nonsignificant findings. Moreover, while 'per-slice analysis was performed treating each slice as independent', it is clear that slices from each patient were actually intercorrelated to some extent. Please seek assistance from a professional statistician to deal with this important point.
2) Why involving novice readers (even a non-radiologist one!) in the first reading session and then having their readings checked by experts. Any automated system is expected to be used by sufficiently experienced, board-certified readers in clinical practice - the statement in the Discussion that the automated protocol could 'reduce tasks requiring the expertise of specialized radiologist the could be replaced by junior radiologist or radiology technicians' is misleading and potentially dangerous in my view.
3) As a minor point, please report the degree of expertise (in terms of years after board certification or of dedicated subspecialty professional activity) of the radiologists expert in abdominal imaging and in radiomics data extraction.
Author Response
Comments 1: While multiple slices were segmented (making up a total of 160 segmentations), 20 patients (of whom 10 with cirrhosis and 10 without cirrhosis) is a very limited sample size, posing a risk of the study being underpowered - this could especially affect statistically nonsignificant findings. Moreover, while 'per-slice analysis was performed treating each slice as independent', it is clear that slices from each patient were actually intercorrelated to some extent. Please seek assistance from a professional statistician to deal with this important point.
Response 1: Thank you for pointing this out. It is true that our patient sample size is limited in that study, particularly when stratifying the analysis by patients with or without cirrhosis. We believe this is addressed by using a better statistical testing approach that can take all slices into account as well as their patient-related correlations. We also note that there are around 40 to 80 slices per patient, leading to a total of around 800 to 1600 slices in the analysis for each modality.
We agree that the per-slice analysis did not consider the correlations between the slices of a same patient, as we mentioned in the assumptions. We now modified the per-slice analysis by applying the aligned rank transform (ART) repeated measures ANOVA, with a main effect of protocol and a main effect of cirrhosis, which is suitable for non-gaussian distributed data, as in our case, and accounts for slice inter-correlation between patients [32].
- Matthew Kay, Lisa A. Elkin, James J. Higgins, & Jacob O. Wobbrock. (2021). mjskay/ARTool: ARTool 0.11.0 (v0.11.0). Zenodo. https://doi.org/10.5281/zenodo.4721941
These modifications provide a more robust statistical framework and address the concerns raised. You can find these modifications in the statistical analysis and results sections.
Finally, we added the oldest per-slice analysis performed treating each slice as independent in the supplementary materials. This provides an additional statistical perspective confirming the enhanced consistency of the annotators in contouring the liver.
Comments 2: Why involving novice readers (even a non-radiologist one!) in the first reading session and then having their readings checked by experts. Any automated system is expected to be used by sufficiently experienced, board-certified readers in clinical practice - the statement in the Discussion that the automated protocol could 'reduce tasks requiring the expertise of specialized radiologist the could be replaced by junior radiologist or radiology technicians' is misleading and potentially dangerous in my view.
Response 2: We understand your comment and concern. Manual segmentation is a time-consuming and labor-intensive task. In this study, we do not focus on segmenting liver lesions, which requires the expertise of a radiologist to detect and characterize lesions. Instead, our focus is on delineating the contours of a shape (liver) while considering certain anatomical details. We believe that this relatively straightforward task can be performed by radiologists in training and radiology technicians, after having received clear instructions outlined in a well-defined and illustrated segmentation protocol. Such an approach could potentially accelerate the generation of datasets available, which are essential for developing advanced computer-assisted analysis.
Moreover, it is important to emphasize that our study includes a radiologist in training with anatomical knowledge and a radiology technician whose daily tasks involve aligning MRI sequences one specific anatomical structures, rather than a non-specialist audience.
Finally, our results demonstrate that the Dice Similarity Coefficient obtained is comparable to that reported in a study comparing two board-certified abdominal radiologists [40]. This reinforces the idea that the task can be reliably performed by radiologists in training and radiology technician without compromising the quality of the segmentations.
- M. Gross, S. Arora, S. Huber, A. S. Kücükkaya, et J. A. Onofrey, « LiverHccSeg: A publicly available multiphasic MRI dataset with liver and HCC tumor segmentations and inter-rater agreement analysis », Data in Brief, vol. 51, p. 109662, déc. 2023, doi: 10.1016/j.dib.2023.109662.
In order to clarify this point, the discussion section (page 10, paragraph 3) has been updated as follows:
"This study focuses on delineating the contours of a shape (liver) rather than segmenting liver lesions, which requires the expertise of a specialized radiologist to detect and characterize lesions. We believe that this relatively straightforward task can be performed by radiologists in training and radiology technicians who have a certain level of anatomical knowledge and after having received a well-defined and illustrative segmentation protocol. Such an approach could facilitate the segmentation process and reduce tasks that require the expertise of a specialized radiologist.”
Comments 3: As a minor point, please report the degree of expertise (in terms of years after board certification or of dedicated subspecialty professional activity) of the radiologist’s expert in abdominal imaging and in radiomics data extraction.
Response 3: Thank you for this comment. We address this point in the materials and methods section (page 3, paragraph 2), cited below:
“All segmentations were reviewed and corrected in consensus among two expert radiologists (C.D. and N.V.V., with 25 and 8 years of expertise in abdominal imaging, respectively) and a radiologist specializing in radiomics data extraction (M.J., with 8 years of expertise in this domain) in order to develop a segmentation protocol.”
Reviewer 2 Report
Comments and Suggestions for Authors
The authors proposed a segmentation protocol for MRI liver scans and found that individual discrepancies decreased when using the protocol. Below are some comments regarding this paper:
1. The pre-protocol individual discrepancy analysis includes only two annotators, which may be insufficient to generalize the findings. Please consider adding more annotators to strengthen this analysis.
2. Please consider moving the description of the proposed protocol to the Materials and Methods section for clarity and better organization.
3. Please describe the approach used to reduce discrepancies between experts in determining the segmentation ground truth accroding to the proposed protocol.
4. It would be helpful to illustrate ground truth segmentation results in Figures 2–5 to enhance the visual clarity of your findings.
Author Response
Comments 1: The pre-protocol individual discrepancy analysis includes only two annotators, which may be insufficient to generalize the findings. Please consider adding more annotators to strengthen this analysis.
Response 1: Thank you for this comment. One of the initial objectives of this study was to assess whether the inter-annotator agreement between a radiologist in training and a radiology technician could match the results of a previous study comparing two board-certified radiologists [40]. The present study was designed following the above-mentioned study’s design, which also included two annotators. Including additional annotators would require recruiting new participants and redoing all statistical analyses from the beginning, which is currently not feasible given to the delay of resubmission (14 days).
Moreover, beyond the analysis of inter-annotator variability itself, a primary objective of this study is to demonstrate that implementing a segmentation protocol significantly improves annotation quality (essential for advanced computer-assisted analysis). The improvement in annotations before and after applying the segmentation protocol has been demonstrated, therefore, we believe that adding additional annotators would not significantly impact the primary outcome’s conclusion.
- M. Gross, S. Arora, S. Huber, A. S. Kücükkaya, et J. A. Onofrey, « LiverHccSeg: A publicly available multiphasic MRI dataset with liver and HCC tumor segmentations and inter-rater agreement analysis », Data in Brief, vol. 51, p. 109662, déc. 2023, doi: 10.1016/j.dib.2023.109662.
We decided to clarify this point in the limitations of the study in the discussion section (page 11, paragraph 2) cited below:
“Moreover, the present study was design following a previous study comparing two board-certified radiologists [40], which is why we chose to include only two annotators. However, beyond the analysis of inter-annotator variability itself, the primary objective demonstrated in our study is that implementing a segmentation protocol significantly improves annotation quality, which is essential for advanced computer-assisted analysis.”
Comments 2. Please consider moving the description of the proposed protocol to the Materials and Methods section for clarity and better organization.
Response 2: Thank you for this suggestion. We consider the development of the segmentation protocol to be a standalone section of our work. To maintain the conciseness of the materials and methods chapter, we chose to summarize the key elements of the protocol in a single paragraph while providing more detailed information in Appendix A. The aim of this approach is to provide interested researchers the possibility to access easily to the segmentation protocol separately, allowing them to apply it in their studies. However, if necessary, Appendix A can be integrated directly into the methodology section.
Comments 3. Please describe the approach used to reduce discrepancies between experts in determining the segmentation ground truth according to the proposed protocol.
Response 3: Thank you for pointing this out. The ground truth was defined as the segmentation corrected by the radiology experts. To reduce discrepancies after the first round of segmentation, we identified specific regions of the liver where the differences between the two annotators were most significant, using the expert segmentation as the reference. For instance, the hepatic hilum, vascular and ligamentous structures were highlighted as key areas of discrepancies. Additionally, clear instructions were provided to the annotators for specific cases, such as those involving multi-part liver parenchyma, respiratory motion artifacts and ascites.
According to your comment, we clarified the materials and methods section (page 3, paragraph 2) as follows:
“All segmentations were reviewed and corrected in consensus among two expert radiologists (C.D. and N.V.V., with 25 and 8 years of expertise in abdominal imaging, respectively) and a radiologist specializing in radiomics data extraction (M.J., with 8 years of expertise in this domain) in order to develop a segmentation protocol. To reduce discrepancies after the first round, specific regions of the liver where the differences between the two annotators were most significant have been analyzed, using the corrected segmentations by the experts as the reference standard. Key areas of discrepancy included the hepatic hilum, vascular structures, and ligamentous regions. Additionally, clear instructions were provided to the annotators for specific cases, such as those involving multi-part liver parenchyma, respiratory motion artifacts and ascites. The segmentation protocol was developed to standardize the process and enhance consistency between annotators."
Comments 4. It would be helpful to illustrate ground truth segmentation results in Figures 2–5 to enhance the visual clarity of your findings
Response 4: Thank you for emphasizing this point. We have modified Figures 2 to 4 and their legends to include an illustration of the ground truth. Figure 5 was not modified as it was intended to illustrate the issue of a missing slice, and we believe that including the ground truth in this figure would not help to enhance clarity.
Round 2
Reviewer 1 Report
Comments and Suggestions for Authors
Thank you for your reply.
Reviewer 2 Report
Comments and Suggestions for Authors
The authors have successfully addressed my previous comments. There are no further comments.